# Predictors of Changes in Physical Activity and Sedentary Behavior during the COVID-19 Pandemic in a Turkish Migrant Cohort in Germany

**DOI:** 10.3390/ijerph18189682

**Published:** 2021-09-14

**Authors:** Lilian Krist, Christina Dornquast, Thomas Reinhold, Katja Icke, Ina Danquah, Stefan N. Willich, Heiko Becher, Thomas Keil

**Affiliations:** 1Institute of Social Medicine, Epidemiology and Health Economics, Charité-Universitätsmedizin Berlin, 10117 Berlin, Germany; christina.dornquast@web.de (C.D.); thomas.reinhold@charite.de (T.R.); katja.icke@charite.de (K.I.); ina.danquah@uni-heidelberg.de (I.D.); stefan.willich@charite.de (S.N.W.); thomas.keil@charite.de (T.K.); 2Institute of Global Health (HIGH), University Hospital Heidelberg, 69120 Heidelberg, Germany; 3Institute for Medical Biometry and Epidemiology, University Medical Center Hamburg-Eppendorf, 20251 Hamburg, Germany; h.becher@uke.de; 4Institute of Clinical Epidemiology and Biometry, University of Würzburg, 97070 Würzburg, Germany; 5State Institute of Health, Bavarian Health and Food Safety Authority, 97688 Bad Kissingen, Germany

**Keywords:** physical activity, sedentary behavior, COVID-19, migrants, public health

## Abstract

The new coronavirus (COVID-19) pandemic and the resulting response measures have led to severe limitations of people’s exercise possibilities with diminished physical activity (PA) and increased sedentary behavior (SB). Since for migrant groups in Germany, no data is available, this study aimed to investigate factors associated with changes in PA and SB in a sample of Turkish descent. Participants of a prospective cohort study (adults of Turkish descent, living in Berlin, Germany) completed a questionnaire regarding COVID-19 related topics including PA and SB since February 2020. Changes in PA and SB were described, and sociodemographic, migrant-related, and health-related predictors of PA decrease and SB increase were determined using multivariable regression analyses. Of 106 participants, 69% reported a decline of PA, 36% reported an increase in SB. PA decrease and SB increase seemed to be associated with inactivity before the pandemic as well as with the female sex. SB increase appeared to be additionally associated with educational level and BMI. The COVID-19 pandemic and the response measures had persistent detrimental effects on this migrant population. Since sufficient PA before the pandemic had the strongest association with maintaining PA and SB during the crisis, the German government and public health professionals should prioritize PA promotion in this vulnerable group.

## 1. Introduction

In November 2019, the first cases of the new coronavirus Sars-CoV-2 emerged in China. From there, the virus and the coronavirus disease (COVID-19) spread all over the world. In March 2020, the World Health Organization (WHO) declared the COVID-19 outbreak a global pandemic [1]. As of July 2021, more than 180 million cases of COVID-19 had been confirmed by WHO including more than 3.9 million deaths [2].

Many countries have imposed response measures, mostly called ‘lockdowns’, to contain the spread of the virus including social distancing, home confinement, or closure of public spaces [3]. In Germany, the first case of COVID-19 was detected on 27 January 2020 [4]. As of 22 March 2020, a nationwide lockdown was implemented including the closure of schools, restaurants, bars, sports facilities, and other measures for social distancing such as working from home, wearing masks, prohibition of mass events, as well as the appeal to stay at home whenever possible [5]. These measures were gradually lifted at the end of May 2020 but had to be reimposed in November–December 2020 during the second wave of the pandemic in Germany [6].

These lockdown measures have influenced people’s work, education, travel, and recreation, and subsequently, their levels of physical activity (PA) and sedentary behaviors (SB) all over the world, as two recent reviews show [7,8]. Research shows that physical activity has many positive effects on physical and mental health and can lead to higher social contentedness [9,10,11]. Additionally, it can help to overcome stress, anxiety, and depressive symptoms, all symptoms that were reported by many persons due to the pandemic [12,13,14]. Sedentary behavior, on the other hand, is associated with an increased risk of musculoskeletal, metabolic, and cardiovascular diseases [15,16,17].

Although exercising alone or with one accompanying person was always allowed in Germany, fitness centers, community sports grounds, public swimming pools, and outdoor leisure facilities were closed. Since people were advised to work from home and to stay at home whenever possible, active transport was reduced as well, while the time spent sitting at home increased. As a consequence, many people reduced their activities, especially persons with little children, older adults, as well as persons who did not cope well with the pandemic-induced stress [18,19,20,21]. Less is known, however, about coping strategies among migrant groups and if PA and SB changes occurred comparably. There is a study from Australia that investigated coping strategies among migrants during the pandemic, but without taking physical activity into account [22]. A recent study investigating a Turkish migrant sample showed that PA and SB changes depended, among other things, on migration-related factors like citizenship, and the preferred language when completing a questionnaire; however, these results were reported before the COVID-19 pandemic [23].

From a public health perspective, it is important to better understand the underlying mechanisms of PA and SB changes to be able to develop public health strategies for specific population groups that are at higher risk of this undesirable behavior, e.g., due to their living situation, work environment, or language barriers. In Germany, persons with Turkish background have formed the largest migrant group (currently 2.82 million) since in the 1960s and 1970s, when Germany recruited so-called “guest workers” from predominantly Southern Europe and the Mediterranean region [24].

The aim of our study was therefore to describe PA and SB changes during the COVID-19 pandemic in a sample of adults of Turkish descent living in Berlin, Germany, and to investigate sociodemographic, health-related, and migration-related factors associated with those changes.

## 2. Materials and Methods

### 2.1. Study Design

This present study is a cohort study among adults of Turkish descent living in the inner-city districts of Berlin. Three assessment points took place in 2011–2012, 2018–2019, and 2020 during the COVID-19 pandemic. The baseline assessment was part of the pretest phase of the German National Cohort (NAKO) intending to evaluate different recruitment strategies (a register-based and a network approach) among persons of Turkish descent using an onomastic procedure. Details of the recruitment are described by Reiss et al. [25].

### 2.2. Participants and Enrollment

For the baseline assessment, all eligible persons were invited to the study center. Inclusion criteria were having Turkish descent and an age between 20 and 69. Persons who were willing to participate completed a questionnaire and underwent medical examination (measurement of body height and weight, a blood pressure test, had a blood sample taken). After 6 to 7 years (between May 2018 and July 2019), a self-administered questionnaire was sent out to all participants who did not refuse to be recontacted (*n* = 557). Participants were asked about health status and behavior, health care utilization, among other questions. A description of the follow-up recruitment has been provided by Krist et al. [26]. The 3rd assessment (2nd follow-up) took place during the COVID-19 pandemic between July and December 2020. A questionnaire, originally used in the German National Cohort (NAKO) [14], including COVID-19 related topics (own infection, living situation, social distancing measures, health behavior, mood since February 2020, the start of the pandemic in Germany) was sent out to all subjects who had not refused to be recontacted since the 1st follow-up (*n* = 377). The 1st invitation letter was sent out in July, the 1st reminder in August, and the 2nd reminder in November 2020 (Figure 1).

Throughout all assessment points, bilingual study material was used to increase participation. The study was approved by the ethical review committee of the Charité—Universitätsmedizin Berlin, Germany, and registered at the German Clinical Trials Register under the registration number DRKS00013545, 08.01.2018. Written informed consent was obtained from all participants.

### 2.3. Measures

#### 2.3.1. Changes in Physical Activity (PA) and Sedentary Behavior (SB)—Outcome

Changes in physical activity, defined as moderate to vigorous activity, during the COVID pandemic were assessed asking for changes in five different settings: work (working at home counted as working, e.g., carrying loads, walking), home, leisure time, sport, and transport. Another question asked on SB, defined as any activity performed while sitting. For each PA setting as well as for SB, participants could choose between ‘much less than before the pandemic’, ‘somewhat less’, ‘no change’, ‘somewhat more’, ‘much more’. Those five categories were dichotomized into ‘less PA’ and ‘equal or more PA’ for physical activity and into ‘more SB’ and ‘equal or less SB’ for sedentary behaviors.

#### 2.3.2. Physical Activity as a Covariate

PA was also assessed at 1st follow-up asking for frequency and duration of PA per week. PA minutes were then calculated and dichotomized into at least 150 min or less than 150 min of PA per week according to WHO recommendations [27]. The created variable (‘WHO recommendations fulfilled yes/no’) was used as a covariate in all analyses.

#### 2.3.3. Sociodemographic Variables

We included age, sex, educational level (assessed at baseline), and employment status (assessed at 2nd follow-up) as sociodemographic variables. Education was assessed as years of education, school type, and country and categorized into <10 years, 10–12 years, and >12 years, taking the Turkish schooling reform into account [28]. We obtained harmonized categories of formal education attained in Turkey and/or Germany. Employment status was assessed as a dichotomous variable yes/no.

#### 2.3.4. Migration-Related Variables

As the first migration-related variable, own migration experience (assessed at baseline) was included. Participants who were born in Turkey or another country were categorized into the group with their own migration experience, while participants who were born in Germany were defined as the group without migration experience.

Second, the language skills of the participants were included using the chosen language of the questionnaire (German or Turkish) since lots of the participants reported two mother tongues (Turkish and German).

#### 2.3.5. Health-Related Variables

As health-related variables, body mass index (BMI), smoking behavior, subjective health status, and mood were included. Body height and weight were measured at baseline by trained study personnel using a calibrated integrated measurement station (SECA model 764, Seca^®^, Hamburg, Germany). From these measurements, BMI was calculated as weight over height squared in kg/m^2^ and categorized into normal weight (BMI 18.5 to < 25.0 kg/m^2^), overweight (BMI 25.0 to <30.0 kg/m^2^), and obesity (BMI ≥ 30.0 kg/m^2^). (There was no participant with a BMI below 18.5 kg/m^2^.) For the analyses in this manuscript, the BMI of the 1st follow-up was used, calculated using measured height at baseline and self-reported weight at 1st follow-up.

Smoking status was assessed at 2nd follow-up and categorized into smoker (regular smoking), ex-smoker, and never-smoker.

Health status was assessed at 2nd follow-up with the question, ‘How would you describe your health status in general?’ The answers ‘excellent’, ‘very good’, ‘good’, ‘not so good’, and ‘poor’ were dichotomized into ‘good’ (including ‘excellent’, ‘very good’, and ‘good’) and poor (including ‘not so good’ and ‘poor’).

#### 2.3.6. Mental Health Variable

The mood of the participants was assessed at 2nd follow-up as well, using the depression module (PHQ-9) from the Patient Health Questionnaire, a validated 9-item tool for measuring depressive symptoms based on DSM-IV criteria with good test–retest reliability [29]. The questionnaire asks for loss of interest, feelings of depression, tiredness, loss of energy or concentration. Each question had four answer categories: ‘Not at all’, ‘several days’, ‘more than half the days’, and ‘nearly every day’ resulting in a sum score. The scores ranged from 0 to 27, the cut-off of ≥10 points representing moderate to severe levels of depression [30,31].

### 2.4. Statistical Analyses

We used an explorative statistical approach rather than conducting strict hypothesis testing, as it was not the aim to create a comprehensive prediction model. Participants’ characteristics were analyzed using descriptive methods of means and standard deviations for continuous data and absolute and relative frequencies for categorical data. A pairwise correlation analysis was performed to detect relations between the different PA settings that were reported as contingency coefficients (Appendix A). The settings transport, leisure time, and sports had a contingency coefficient of 0.7 indicating a strong correlation and were subsequently combined to the outcome variable for PA change. Since changes in the settings work and home were less nuanced, they were not included in the main analysis, but only presented in the supplement (Appendix A). A multivariable logistic regression analysis was conducted to investigate associations between variables assessed at baseline, 1st, and 2nd follow-up (exposures) and changes of PA (combined settings) and SB during the pandemic (outcomes). For the multivariable analyses, covariates with *p* > 0.2 were removed using backward-stepwise elimination (Wald) (i.e., the least significant covariate was removed one at a time, then the model rerun). The remaining variables of the final model were included in a complete case multivariable logistic regression model. As a measure of goodness-of-fit, we report Nagelkerke’s R-squared values. Included variables were not correlated with each other. Results of the multivariable regression analysis were presented as odds ratios (OR) with 95% confidence intervals (CI). The analyses were performed using SPSS Statistics for Windows (25.0.0.1, IBM Corp., Armonk, NY, USA).

## 3. Results

Out of 377 persons who were contacted between July and December 2020 for the 2nd follow-up of our cohort study, 106 completed the questionnaire (44 after the first invitation, 57 after the first reminder, and 5 after the second reminder). Two hundred and sixty-three persons did not answer, four persons actively declined to participate, three persons could not be reached due to a wrong address, and one person had died.

Among the sample, mean age ± standard deviation (SD) was 53.9 ± 11.8 years, 58% were women, and 82% had their own migration experience. Before the COVID-19 pandemic, 22% were sufficiently active with at least 150 min of moderate PA per week. All participants’ characteristics are presented in Table 1.

Among all participants, 68.9% reported a decline of PA in any of the five settings (55.6% among the formerly active, and 76.6% among the inactive, (Figure 3a)). The decline was the strongest in sports (55.7%), followed by leisure time PA (43.1%), active transport (42.3%), activities at work (28.3%), and at home (8.1%) (Figure 2). Regarding SB, 36% reported an increase since the start of the pandemic (11.8% among the active, and 46.7% among the inactive) (Figure 2 and Figure 3b).

When stratifying the participants for having fulfilled the WHO recommendations regarding PA (at least 150 min per week at least moderately active) before COVID-19, changes in PA and SB differed between the strata (Figure 3a,b).

### Regression Analyses

Univariable regression analysis did not show any statistically significant association between PA changes and the included predictor variables; SB changes were associated with inactivity in the past (Table 2 and Table 3).

Multivariable regression analyses revealed that female sex and low PA in the past seemed to be associated with a decreased PA during the pandemic. Women had 70% higher odds than men to engage in less PA during the pandemic (OR 0.3 (0.1;0.9)), and persons who did not fulfill WHO recommendations at 1st follow-up were more than 6 times more likely to report a decrease in PA during the pandemic than their active counterparts (OR (95%CI): 6.2 (1.7;22.6)) (Table 2). Higher SB appeared to be associated with female sex, high BMI, and inactivity in the past. Women had 70% higher odds of sitting more than men (OR 0.3 (0.1;0.9)). Persons who were obese compared to normal or overweight persons were three times, and persons who were not physically active in the past 19 times more likely to sit more during the pandemic (OR: 3.3 (1.0;10.4), and 19.3 (2.2;170.0), respectively) (Table 3).

## 4. Discussion

The present study aimed to investigate changes in PA and SB among a sample of adults of Turkish descent in Berlin, Germany, during the COVID-19 pandemic, and to determine predictors of these changes.

More than two-thirds of the participants reported a decline in PA since the start of the pandemic, and more than one-third reported an increase in SB. Similar proportions have been suggested by other authors ranging from 40–70%, however, only for non-migrant populations [32,33]. The most important PA decrease was observed in sports, leisure time, and transport, while PA at work decreased moderately and household activities decreased only to a small extent. Since sports facilities were closed and people were advised to stay at home, as well as to work from home, it is comprehensible that PA decreased the most in these three settings and remained almost stable for household activities. A German study supports this assumption by showing that 41% of the participants reported closed sports infrastructure as a reason for reduced PA [21]. Another study from Germany reports similar results for a non-migrant study sample as our study regarding different settings with the highest decrease in sports, leisure time, and work PA, but unchanged household activities [34]. Although our results did not show an association with age, it seems that in younger individuals PA decrease was less nuanced than in older ones. Huber et al. described a decrease of PA in 45% of a sample with a mean age of 23 years [35], and in a German sample of children aged 4 to 17, almost no change of PA was shown; however, screen time increased to a very large extent [36].

The second main finding from our study regards predictors of PA and SB changes. Persons who did not meet the recommendations of being moderately active before the pandemic for at least 150 min per week were more than six times more likely to report decreased PA during the pandemic. This was in line with a study from Italy reporting sufficient activity before the pandemic as one of the predictors to stay active during the pandemic [33]. In contrast, two studies showed a reduction among persons who were active before the pandemic, but not among the inactive [37,38], and a study from the UK reported a reciprocal association between the intensity of PA performed before and its reduction during the pandemic [39]. As reported by several other studies, female sex was also associated with decreased PA in our sample, however, depressive symptoms and health status were not associated, even though 30% of the participants reported depressive symptoms [37,40,41]. Age was not associated with decreased PA either, and neither was education, which is partly in line with another German study that did show an association of higher education with being active during the pandemic, but no association of age and sex, either [18]. 

Time spent sitting increased in 36% of the participants. This proportion is in the range of 30–50% reported by other authors [33,37,42,43]. A Chinese study reported 67% of persons indicating increased SB; however, the two-to-three month home confinement in China was one of the strictest in the world [44].

Predictors for increased SB were female sex, being obese compared to normal or overweight, and not meeting PA recommendations before the pandemic. This is in contrast to several other studies where SB decrease was rather an overall phenomenon and not specific for some subgroups [37,43,45]. Among the studies that reported associated factors with SB increase, factors were changes in working situation (working from home or having lost a job) [46], male sex, and non-smoking [47]. While the previous results of this cohort study showed that German citizenship and German as preferred questionnaire language were associated with a positive PA trajectory [23], in the present study none of the migration-related variables was associated with decreased PA or increased SB. Since the results of this study were comparable with other studies conducted in Germany, at least for Berlin there does not seem to be a large difference between the population of origin and persons with a Turkish background.

### 4.1. Strengths and Limitations

To the best of our knowledge, this is the first study investigating PA and SB changes in persons with a migrant background in Germany during the COVID-19 pandemic. The proportion of studies that focus on migrants is still very small. Especially during a crisis such as the COVID-19 pandemic, it is important to collect data of all relevant subpopulations to be able to react adequately and to inform these subgroups about risks and respective prevention measures during the crisis. Migrants often live in a precarious housing situation, are employed at workplaces where working from home is not possible, or have language difficulties which are barriers to a healthy lifestyle and also risk factors for a coronavirus infection [48]. Therefore, data like ours on preventive lifestyle behaviors is crucial for better preparedness for future pandemics.

The present study will help to provide information about the health behavior of Turkish migrants in Germany and could thus be used to address this population group more individually. Another strength of our study was that PA behavior before the pandemic was assessed before and not as recall during the already existing pandemic. Third, data collection started six months after the start of the response measures including the summer with only a few restrictions and the autumn/winter where a nationwide lockdown was reimposed. These data represent therefore a more realistic and long-term effect of the pandemic and the restrictions on changes in PA and SB. Apparently, reduction of PA and increase of SB are not only an acute problem during a lockdown but continue even if response measures have been lifted.

Among the limitations of our study, we need to point out that the number of participants decreased considerably after the baseline assessment. Differences between participants and non-respondents may have led to somewhat distorted results. However, since unfavorable health behaviors were more common among non-respondents than among participants, changes in PA and SB were rather under- than overestimated. This assumption is supported by Wunsch et al., who investigated the effects of the pandemic on PA changes as well, performing a non-respondents’ analysis and showing lower PA and higher SB at baseline among the non-respondents compared to the respondents [49]. Another limitation is reduced significance of the study due to the small sample size. A further limitation is the limited comparability of studies from different countries focusing on this topic. While some countries such as China, Italy, or France imposed home confinements and strict curfews, the measures in Belgium, Switzerland, and Germany were less drastic. Lastly, a self-administered questionnaire has its limitations and increases the risk of response bias due to social desirability or recall difficulties, even if the questions were taken from validated questionnaires [50,51].

### 4.2. Implications

Our results emphasize the importance of providing possibilities to maintain PA even during a crisis like the COVID-19 pandemic. A recent review showed evidence of the benefits of exercise programs designed to motivate people during a home confinement, including reducing their feeling of loneliness, e.g., online training programs, exergames, programs for the whole family/household, activities at home, or use of fitness trackers [52]. WHO, the American Heart Association (AHA), and the American College of Sports Medicine (ACSM) offered guidelines regarding activities during home confinement [53,54,55], but these associations are, however, not very well known in Germany. In addition, persons with lower socioeconomic status including a considerable number of migrants with a Turkish background are already less likely to engage in sufficient physical activity [56,57]. Besides financial barriers and the accessibility of sports facilities, gender roles, social expectations, language problems, as well as religious aspects influence the engagement in PA [58]. Exercise promotion in Germany is already very limited in normal times, therefore, after this global health crisis, essential government actions should be the implementation of nationwide exercise programs and information campaigns regarding health benefits of PA and strategies to reduce SB. In addition to general prevention measures, special emphasis should be put on more personalized programs, as proposed by the investigators of the DEDIPAC study [59], targeting vulnerable persons, such as persons with low socioeconomic status, or migrants and their descendants. These could be peer-supported programs promoted in the respective migrant communities [60].

## 5. Conclusions

Our study provides evidence of persistent lifestyle behavioral consequences of the COVID-19 pandemic, demonstrating an adverse effect on PA and SB among a sample of adults with Turkish roots living in Germany. Among this sample, PA was a strong predictor for staying active during this pandemic crisis. These findings highlight the importance of PA promotion efforts, but not only specific to the pandemic situation, where sports facilities were less accessible, and people were staying most of the time at home.

## Figures and Tables

**Figure 1 ijerph-18-09682-f001:**
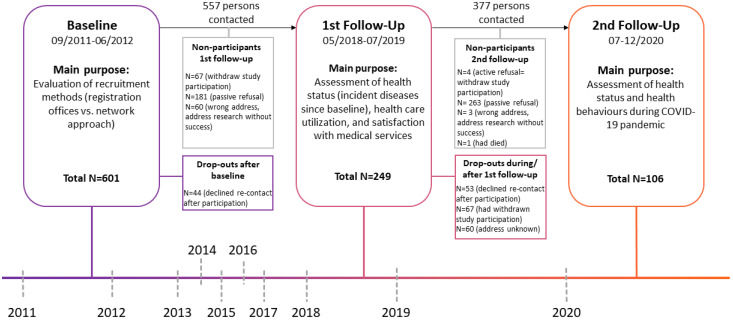
Recruitment process.

**Figure 2 ijerph-18-09682-f002:**
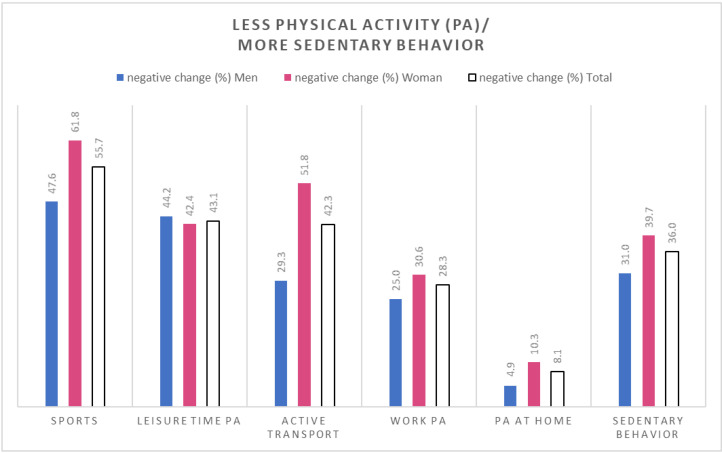
Proportion of participants who reported less physical activity (in five different settings) and more sedentary behavior during the new coronavirus (COVID-19) pandemic (since February 2020) than before.

**Figure 3 ijerph-18-09682-f003:**
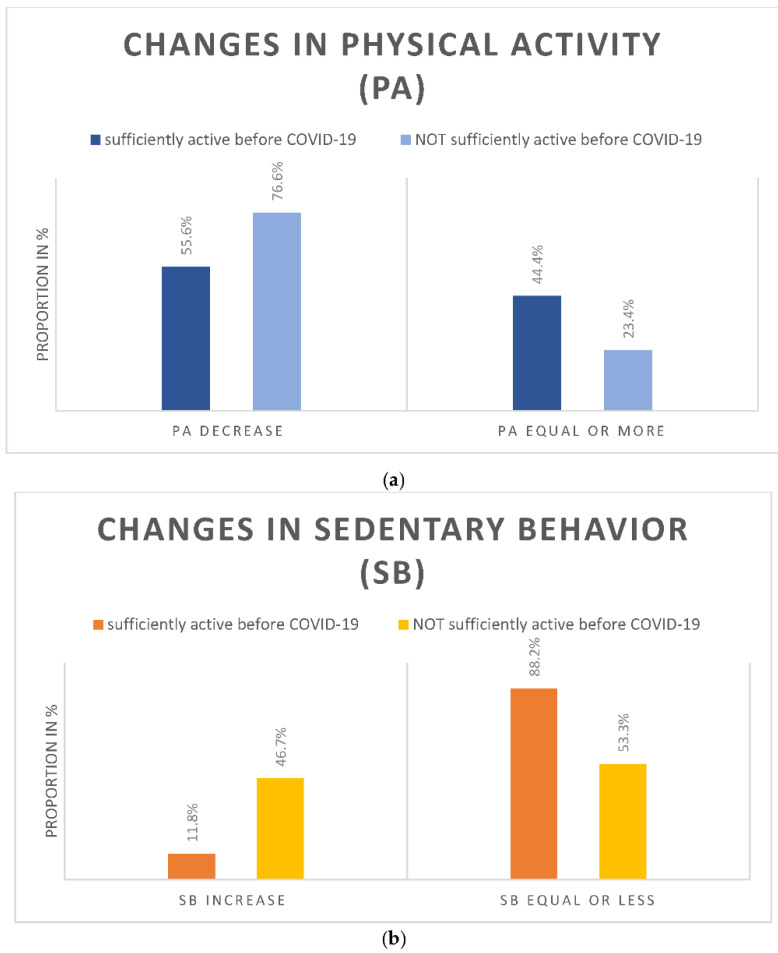
The proportion of participants reporting (**a**) changes in PA (PA decrease: decrease in any of the 5 PA settings); (**b**) changes in SB. Changes are stratified for having fulfilled WHO recommendations regarding PA before COVID-19.

**Table 1 ijerph-18-09682-t001:** Characteristics of the study sample.

	Men *N* = 44	Women *N* = 62	Total *N* = 106
	% or mean ± standard deviation		
Sociodemographics			data
Age in years	54.8 ± 10.9	53.2 ± 12.5	53.9 ± 11.8
Educational level ^1^			
<10 years	22.7	41.9	34.0
10–12 years	47.7	33.9	39.6
>12 years	27.3	16.1	20.8
Questionnaire language			
German	61.4	48.1	59.4
Turkish	38.6	41.9	40.6
Own migration experience ^1^	88.6	77.8	82.0
Employed (half or full time)	54.8	38.3	45.1
Health-related variables			
Body mass index (BMI) ^2^	29.7 ± 4.7	29.3 ± 5.3	29.5 ± 5.1
Normal weight (18.5 to <25.0 kg/m^2^)	9.7	23.2	18.4
Overweight (25.0 to <30.0 kg/m^2^)	41.9	33.9	36.8
Obesity (≥30.0 kg/m^2^)	48.4	42.9	44.8
Smoking behavior			
Smoker	34.1	30.0	31.7
Ex-smoker	22.7	28.3	26.0
Never smoker	43.2	41.7	42.3
Physical activity before COVID-19 pandemic ^2^			
At least 150 min/week of moderate physical activity	17.6	25.0	22.0
Health status, self-reported			
Good (including good, very good, excellent)	86.4	71.0	77.4
Poor (including less good, poor)	13.6	29.0	22.6
Depressive symptoms (≥10 points on PHQ-9 scale ^3^)	25.0	34.5	30.5

^1^ Variables assessed at baseline (2011/2012); ^2^ variables assessed at 1st follow-up (2018/2019); ^3^ PHQ-9: patient health questionnaire depression.

**Table 2 ijerph-18-09682-t002:** Univariable and multivariable regression analysis for the outcome less physical activity during the COVID-19 pandemic.

Predictors		Less Physical Activity in Transport, Leisure Time, and/or Sports			
		Univariable		Multivariable ^4^	
	*n*	OR (95% CI)	*p*	OR (95% CI)	*p*
Sociodemographics					
Age per year	103	1.0 (1.0;1.1)	0.383	-	
Sex ^1^					
Male vs. female	103	0.5 (0.2;1.1)	0.096	0.3 (0.1;0.9)	0.032
Educational level ^1^					
<10 years	100	Ref		-	-
10–12 years		2.0 (0.8;5.2)	0.141	-	-
>12 years		1.8 (0.6;5.5)	0.300	-	-
Employed					
No vs. yes	99	0.8 (0.4;1.9)	0.636	-	-
Migration-related variables					
Questionnaire language					
Turkish vs. German	103	1.2 (0.5;2.8)	0.678	-	-
Own migration experience ^1^					
No vs. yes	87	2.7 (0.7;10.2)	0.143	-	-
Health-related variables					
Obesity vs. normal/overweight ^2^	85	1.3 (0.5;3.2)	0.616	2.0 (0.6;6.9)	0.254
Never/ex-smokers vs. smokers	101	0.6 (0.2;1.4)	0.214	0.4 (0.1;1.4)	0.146
<150 min PA per week vs. ≥150 min PA per week (WHO recommendations)—before COVID-19 pandemic ^2^	80	2.6 (0.9;7.8)	0.072	6.2 (1.7;22.6)	0.006
Subjective health status: poor (not so good or poor) vs. good (good, very good, or excellent)	103	0.8 (0.3;2.1)	0.633	0.3 (0.1;1.3)	0.115
Depressive symptoms (≥10 vs. <10 on PHQ-9 scale ^3^)	95	1.4 (0.5;3.7)	0.487	-	-

^1^ Variables assessed at baseline (2011/2012); ^2^ variables assessed at 1st follow-up (2018/2019); ^3^ PHQ-9: patient health questionnaire depression. ^4^ Complete case analysis with remaining variables of the final model of Wald’s backward elimination), *n* = 74; Nagelkerke’s R-squared: 0.2.

**Table 3 ijerph-18-09682-t003:** Univariable and multivariable regression analysis for the outcome more sedentary behavior during the COVID-19 pandemic.

Predictors		More Sedentary Behavior			
		Univariable		Multivariable ^4^	
	*n*	OR (95% CI)	*p*	OR (95% CI)	*p*
Sociodemographics					
Age per year	100	1.0 (1.0;1.0)	0.467	-	-
Sex ^1^					
Male vs. female	100	0.7 (0.3;1.6)	0.371	0.3 (0.1;0.9)	0.040
Educational level ^1^					
<10 years	100	Ref		-	-
10–12 years		1.5 (0.6;4.0)	0.393	-	-
>12 years		1.3 (0.4;4.2)	0.610	-	-
Employed					
No vs. yes	96	1.3 (0.5;3.0)	0.575	-	-
Migration-related variables					
Questionnaire language					
Turkish vs. German	100	1.2 (0.5;2.7)	0.682	-	-
Own migration experience ^1^					
No vs. yes	84	1.0 (0.3;3.0)	0.957	-	-
Health-related variables					
Obesity vs. normal/overweight ^2^	82	1.5 (0.6;3.6)	0.398	3.3 (1.0;10.4)	0.043
Never/ex-smokers vs. smokers	98	1.0 (0.4;2.4)	0.993	-	-
<150 min PA per week vs. ≥150 min PA per week (WHO recommendations)—before COVID-19 pandemic ^2^	77	6.6 (1.4;31.2)	0.009	19.3 (2.2;170.0)	0.008
Subjective health status: poor (not so good or poor) vs. good (good, very good, or excellent)	100	1.3 (0.5;3.4)	0.587	-	-
Depressive symptoms (≥10 vs. <10 on PHQ-9 scale ^3^)	92	2.4 (0.9;5.9)	0.062	-	-

^1^ Variables assessed at baseline (2011/2012); ^2^ variables assessed at 1st follow-up (2018/2019); ^3^ PHQ-9: patient health questionnaire depression. ^4^ Complete case analysis with remaining variables of the final model of Wald’s backward elimination), *n* = 72; Nagelkerke’s R-squared: 0.3.

## Data Availability

The datasets used for this study are available from the corresponding author on reasonable request.

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
