# Peer review of "Predictors of Changes in Physical Activity and Sedentary Behavior during the COVID-19 Pandemic in a Turkish Migrant Cohort in Germany"

_ijerph, 2021, doi:10.3390/ijerph18189682_

Round 1
Reviewer 1 Report
1 My most concern for this manuscript is the sample size. For regression analysis, it is suggested at least 10 observations per variable. In this case, at least 110 respondents, while in the manuscript only 67 were analyzed. The small sample size decreases statistical power. In the discussion section, it seems authors have already known about the issue. My suggestion is maybe the authors can try other statistic methods, or reduce the number of variables.
Also, as the authors used regression model, it confused me of Table 2 and Table 3, what “univariable regression” used for? It is not stated in the paragraph.
The goodness of fit of the models is important information, but they were not mentioned.
Whether the regression assumptions were met? Please state it clear in Methods and Results sections.
2 Besides, my major concern is about the theoretical framework of this study. How authors decided on the predictors (independent variables)? Other variables may also affect conducting physical activity during the pandemic, such as the surroundings of the living place, whether there are suitable green spaces for exercise in the neighborhood; and the family members, do they have little children need to care about; if they need to walk dogs every day. There are many relevant papers.
The question is: Whether the study missed some important variables, resulting in insufficient explanatory power? I’m not sure, as goodness of fit was missed.
3 It is an interesting topic about migrants, but this point was not emphasized in the text. I think maybe for migrants, social cohesion may be an obstacle for them.
The writing is good to understand.
Minors:
L168-172, in the text, it is “between July and December,” but in Figure 1, it shows “08-12/2020”
Reviewer 2 Report
The manuscript entitled "Predictors of changes in physical activity and sedentary behavior during the COVID-19 pandemic in a Turkish migrant cohort in Germany" has an important topic, but the manuscript needs to revise, and it needs an extensive English style edit.
I have the following comments for the authors:
- The authors well introduced the research problem and showed several previous studies on the topic. However, it's not clear why the Turkish migrant groups are important since COVID-19 changed behavior in all levels of society and finding on physical activity (PA) and sedentary behavior (SB) are consistent in many cases.
- Since the authors showed that it's a part of a longitudinal study, I recommend providing information about previous literature about Turkish migrant's health behavior, including PA and SB
- I recommend reconstructing the "2.2. and 2.3. subchapters. It would be ideal to name them as "measures" and put every information about the measures. There is inconsistency in it as well. For example, it is unclear how PA was assessed (line 140) since it was a subchapter about PA already.
- Use different subchapters for the other measures. In this way, it will be easier to follow.
- In subchapter 2.1. (Study design and participants) there is no information about the participants, only the procedure. Thus, this title is misleading. I recommend relocating the first part of the result section here (Line 169-174+figure 1).
- Include reliability of the Patient Health Questionnaire. What were the answer categories of the scale? I also recommend using a different subchapter to introduce the scale.
- I recommend renaming "PA as much or more" signs. It's hard to understand what the authors meant by the name.
- Please describe what "PA at work" means (line 262)
- The self-administered questionnaire has its limitations. Please add it to the limitation part.
Round 2
Reviewer 1 Report
(1) As authors stated the aim of this study was not to predict, the text must follow this idea throughout the manuscript.
For example:
L11-12 in the abstract: “… study aimed to investigate predictors of changes in PA and SB in a sample of Turkish descent.” This sentence needs to be revised.
L30-31 also in the abstract: “Since sufficient PA before the pandemic was the strongest predictor for maintaining PA and SB during the crisis”
L282-283: “Multivariable regression analyses revealed that the only predictors for a decreased PA during the pandemic were female sex and low physical activity in the past.”
(2) L27-28: “PA decrease and SB increase were associated with inactivity before the pandemic as well as female sex.” As the sample size was relatively small, the results should be reported tentatively. For this, and for the whole text in the results section.
(3) Although this manuscript is easy to understand, there are small grammatical errors, please check and revise.
(4) L291-292: “…persons who were not physically active in their past 19 times more likely to sit more during the pandemic…” The odds is very high, please carefully check the data (the sample size for each subgroup), can it be a bias?
(5) L402: 4.2. Implications, authors should link the implications with the findings and the subjects (Turkish migrants), and make more specific suggestions.
Reviewer 2 Report
The manuscript is well corrected. Thank you for the author's contirbution.
Author Response
Thank you very much!